# The Effects of Friction and Temperature in the Chemical–Mechanical Planarization Process

**DOI:** 10.3390/ma16072550

**Published:** 2023-03-23

**Authors:** Filip Ilie, Ileana-Liliana Minea, Constantin Daniel Cotici, Andrei-Florin Hristache

**Affiliations:** 1Department of Machine Elements and Tribology, Polytechnic University of Bucharest, Spl. Independentei 313, 060042 Bucharest, Romania; 2Department of Biotechnical Systems, Polytechnic University of Bucharest, Spl. Independentei 313, 060042 Bucharest, Romania

**Keywords:** selective transfer, CMP process, SiO_2_ slurry, abrasives particles, friction energy, thermal effect

## Abstract

Chemical–mechanical planarization (CMP) represents the preferred technology in which both chemical and mechanical interactions are combined to achieve global planarization/polishing of wafer surfaces (wafer patterns from metal with a selective layer, in this paper). CMP is a complex process of material removal process by friction, which interferes with numerous mechanical and chemical parameters. Compared with chemical parameters, mechanical parameters have a greater influence on the material removal rate (*MRR*). The mechanical parameters manifest by friction force (*F_f_*) and heat generated by friction in the CMP process. The *F_f_* can be estimated by its monitoring in the CMP process, and process temperature is obtained with help of an infrared rays (IR) sensor. Both the *F_f_* and the *MRR* increase by introducing colloidal silica (SiO_2_) as an abrasive into the selective layer CMP slurry. The calculated wafer non-uniformity (*WNU*) was correlated with the friction coefficient (*COF*). The control of *F_f_* and of the slurry stability is important to maintain a good quality of planarization with optimal results, because *F_f_* participates in mechanical abrasion, and large *F_f_* may generate defects on the wafer surface. Additionally, the temperature generated by the *F_f_* increases as the SiO_2_ concentration increases. The *MRR* of the selective layer into the CMP slurry showed a non-linear (Prestonian) behavior, useful not only to improve the planarization level but to improve its non-uniformity due to the various pressure distributions. The evaluation of the *F_f_* allowed the calculation of the friction energy (*E_f_*) to highlight the chemical contribution in selective-layer CMP, from which it derived an empirical model for the material removal amount (*MRA*) and validated by the CMP results. With the addition of abrasive nanoparticles into the CMP slurry, their concentration increased and the *MRA* of the selective layer improved; *F_f_* and *MRR* can be increased due to the number of chemisorbed active abrasive nanoparticles by the selective layer. Therefore, a single abrasive was considered to better understand the effect of SiO_2_ concentration as an abrasive and of the *MRR* features depending on abrasive nanoparticle concentration. This paper highlights the correlation between friction and temperature of the SiO_2_ slurry with CMP results, useful to examine the temperature distribution. All the *MRRs* depending on *E_f_* after planarization with various SiO_2_ concentrations had a non-linear characteristic. The obtained results can help in developing a CMP process more effectively.

## 1. Introduction

In a friction pair, a selective transfer can certainly be made if, in the contact area, there exists an adequate lubricant (such as glycerin), a copper-based material (in this case, bronze), relative movement, and favorable energy [1]. By investigating the CMP process of the selective layer, in which copper was the predominant element (≥85%), it was found that under normal conditions, its surface is oxidized. The rates of oxide removal differ depending on the applied load, the removal depth, and the slurry used. There is extensive research on the CMP process, but for a better understanding of the mechanical and tribochemical phenomena, additional studies are needed that appear at the interface of the pad–wafer (selective layer) in the presence of fluid slurry [1,2,3,4,5]. The studies carried out by Ilie [1] and Lee et al. [2] showed that the formation of an oxide film on the selective layer surface requires the use of an oxidizer. However, due to the volatile copper compounds that are formed only at high temperatures, dry etching on the selective layer surface is not practical [6].

The parameters involved in the CMP process are the relative speed between the planarization pad and wafer, abrasive nanoparticle, pressure on the wafer, slurry chemistry, substrate characteristics, and process temperature, respectively. In general, during the CMP process, a metal removal mechanism takes place that depends on the synergy of both mechanical abrasion and chemical dissolution [7]. After the oxidation agent reacts with the copper of the selective layer, a passive film of porous metal oxide forms, which can be easily hydrolyzed and removed through the mechanical friction generated between the wafer and the abrasive nanoparticles. The slurry used in selective-layer CMP is composed of abrasive nanoparticles mixed into an aqueous solution that contains an inhibitor, an oxidizer (H_2_O_2_), a complexing agent, and other additives. H_2_O_2_ is a widely used oxidizer for copper CMP [8]; therefore, it can also be used for copper oxidation from the selective layer. The CMP mechanism used in citric-acid-based slurry and the H_2_O_2_ as oxidizer were presented in ref. [9]. Their results show that the copper anodic reaction is provoked by the increase in corrosion potential and, hence, the passivation layer of copper oxide is increased and the copper dissolution is reduced. Additionally, it was specified that by adding a small amount of H_2_O_2_ to the CMP slurry, the *MRR* improved [10]. One later addition of H_2_O_2_ suppressed the *MRR*, while the copper film surface roughness was very much dependent on the H_2_O_2_ concentration [11].

For a selective-layer wafer pattern, the main goal of the CMP is to remove the low deformed areas without quickly chemically oxidizing them [12]. During over-planarization (over-polishing), the selective layer is sensible to deformation; consequently, a corrosion inhibitor should be protected in low areas of the patterned wafer, because the corrosion is fatal for the copper of the selective layer.

The key element in the oxidizer election is its pH, resulting in the metal oxide film; therefore, several studies were carried out in order to understand the selective-layer CMP process [6,10,11,12,13]. These studies showed that the selective-layer *MRR* can be increased by obtaining a soft and porous surface, and the friction is reduced by softening the selective-layer surface. Additionally, regarding the friction force during the CMP of the selective layer, a modification of Preston’s equation was proposed using an in situ quantitative measurement technique [14].

Citric acid is used as a complexing agent because copper ions from the selective layer can be complexed with citrate ions [14], and benzotriazole (*BTA*) is the most used as an inhibitor. *BTA* added in the CMP slurry forms a Cu-BTA protective layer on the selective-layer surface, and it decreases the chemical dissolution rate of selective-layer copper [13]. The chemical and mechanical parameters change the friction characteristics because friction force depends on them. Thus, a change in the functioning conditions or in the slurry content leads to contradictory results. A critical analysis is needed to better understand the complexity of the tribological interaction between the slurry, the planarization head, wafers, the planarization pad, carrier film, and the pad balms. The friction and heat origin in the CMP process is difficult to define and very complex. However, it is certain that the pad’s surface topography, hardness, and material type to be removed, together with the abrasive used, slurry chemistry, contact pressure, relative speed, etc., will influence the tribological phenomena.

Several studies were conducted on the friction force in the CMP process, as well as on the process temperature, because these parameters offer important information about the planarization state. Wei et al. in ref. [13] studied the link between the friction forces and sizes of some abrasive particles, while Kim in ref. [14] analyzed the tribological state during the CMP process, how it influences the pad surface, and its characterization.

To elucidate the copper CMP mechanism, the mechanical parameters of the copper CMP process were analyzed and approached. Lee et al. in ref. [15] analyzed the mechanical aspects of the *MRR* phenomenon, and Levert et al. in ref. [16] researched the oxide’s influence on the friction force in the copper CMP process. Furthermore, they proved the dependence of the friction force on the nature of the oxide film generated by H_2_O_2_ during copper CMP. Instead, Luan et al. [12] and Stojadinovic et al. [17] proved that the copper *MRR* grows as the abrasive’s concentration increases. Additionally, the *MRR* depends on the covered surface by the dispersed abrasives particles, the sizes, and the abrasive particles’ shape [18]. Both at copper CMP and at selective-layer CMP, the physical defects (such as scratching and etching), are related to the friction force during CMP. As a result, they allow an important understanding of the friction characteristics of the selective-layer CMP process. Thus, this paper investigates the friction, thermal effect, and the *MRRs* of the selective-layer CMP process depending on SiO_2_ concentrations in acid slurry (a mixture of an oxidizer, corrosion inhibitor, a complexing agent, and a surfactant) by monitoring the friction force and the stability of the slurry used.

The results of this investigation can help in developing a CMP process more effectively, with an optimal *MRR* to achieve a quality planarization.

## 2. Materials and Methods

The friction and thermal investigation in the CMP process on a selective-layer surface were carried out on a wafer pattern with a selective layer. The wafer pattern from metal was made of OLC45 steel (equivalent AISI 1045), previously coated on one face with a selective layer (thickness of about 500 nm) by friction with the CuSn12T bronze (equivalent CC483K).

The planarization experiments were performed by using a CP-4 CMP planarization (polisher) installation with a polyurethane pad and an acid slurry mixed with an oxidizer, corrosion inhibitor, complexing agent, and surfactant. The CMP planarizer/polisher used was of rotational type, and the pressing normal down-force between the wafer and the planarization pad was achieved with the help of the air pressure from a supply source and was uniformly applied to the wafer carrier. The applied pressure ranged from 0.5 psi to 3.0 psi, and the rotational speed normal at planarization ranged between 0 and 100 rpm. The CP-4 CMP planarizer was properly equipped with in situ monitoring sensors to signal the (detected) friction force and temperature during the CMP process. After being converted and amplified, the signals emitted by sensors were transferred to the CMP analysis software DAQ 21.8 (a data acquisition system). The infrared (IR) sensor used for monitoring the friction force was placed at the polishing head edge. The signals of temperature and friction force were displayed in real-time on a monitor. The dynamic friction force signal was registered with a piezoelectric sensor attached to the planarization head in the form of voltage, and then was amplified and converted. All these signals were analyzed with the help of the CMP analysis software.

## 3. Theoretical Aspects

The measurement points of the process temperature between the wafer and pad were at the contact area edge. Regarding the measured temperature in the CMP process, although not considered the real temperature, the technique used can indirectly indicate the characteristics of the process temperature. Using the America Society of Testing Materials (ASTM F1530-94), standard deviation uniformity can be determined using the wafer non-uniformity (*WNU*), with the relationship:(1)$$WNU={\frac{\sigma }{MRR}}_{avg}\times 100\left(\%\right)$$
where, *σ* is standard deviation and *MRR_avg_* is the average removal rate of the material.

Implicitly, the *WNU* calculation also demonstrates the non-uniformity of the CMP process temperature, so by measuring the temperature at the edge of the wafer-pad contact area, we cannot say with certainty that it represents the real temperature. However, by measuring the temperature at several points along the edge of the wafer-pad contact area with the help sensors, and analyzing the signals emitted and transferred to the data analysis software DAQ 21.8, we can consider with sufficient accuracy that these measurements represent the temperature characteristics of the CMP process.

The *MRR* expresses itself with the Preston equation [19] in the form:*MRR = k∙p∙v_r_*(2)

Equation (2) shows the *MRR* dependence by the Prestonian constant, *k,* the pressure, *p*, and relative speed, *v_r_*. The constant, *k*, includes the slurry characteristics of planarization equipment, pad, and other environmental factors.

Friction energy *E_f_* [20] can be determined using Equation (3) in the CMP process, namely:(3)$$E_{f}=v_{r}\int_{0}^{t}F_{f}dt$$
where: *F_f_* is friction force and *t* is planarization (polishing) time.

The expression for *E_f_* in Equation (3) represents the mechanical contribution in the *MRR* during planarization/polishing by CMP. Although the mechanical contribution has a greater influence on the *MRR* than the chemical one, it should not be ignored when studying the thermal effects in the CMP process. This is also due to the fact that the chemical reaction plays a great role in the CMP process.

Thus, the multiplication between the activation energy (*E_a_*) and the molecular frequency defines the chemical reaction rate. Additionally, with the growth of molecular kinetic energy, which surpasses *E_a_*, the process temperature increases, and the chemical reaction probability is elevated. For this reason, the reactive constant (*K*) from the Arrhenius-type equation [21] becomes larger and is given by Equation (4):*K* = *A*∙exp(−*E_a_*/*RT*),(4)
where *A* is the Arrhenius constant, *E_a_* is the activation energy, *R* is the gas universal constant, respectively, and *T* is temperature.

In the selective-layer CMP process, understanding the *MRR* features depending on nanoparticle concentration could be useful for the *MRR* calculation per particle. According to Equation (2) the *MRR* can be established with Preston’s equation and it depends on *p* and *v_r_*. However, the *MRR* during the CMP process may also be expressed depending on the *E_f_*. Thus, Stojadinovic et al. in ref. [17] presented the *MRR* calculation on a single abrasive (proposed by Tamboli et al. in ref. [22]), considering the *MRR* produced by the planarization pad. Hence, if it is admitted that the abrasive nanoparticles are nearly spherical, their distribution and size can be neglected, and all participate in planarization, then per an abrasive nanoparticle single, the *MRR* calculation equation can be rewritten, thus: (5)$$MRR_{per\;an\;abrasive}=\frac{MRA-MRR_{pad\;only}}{W\Phi_{m}\left(1/6\pi d^{3}\rho \right)},$$
where *MRA* is the material removal amount; *MRR_pad only_* is the *MRR* at an abrasive concentration of 0.0 wt.%; *ρ* is the abrasive density; *d* is the abrasive size; $\Phi_{m}$ is the slurry mass flow during the CMP; and *W* is the fractional weight of the abrasive in the slurry.

In fact, the *E_f_* is a physical representation of the environmental conditions, pressure, relative motion, material characteristics, surface conditions of the chemically etched wafers, and other planarization/polishing conditions. The use of *E_f_* can make it easy to represent mechanical action/abrasion in CMP, considering the planarization/polishing time, and the *MRA* can be calculated with the following empirical equation:(6)$$\;MRA=C_{c}\cdot E_{f}=C_{c}(v_{r}\int_{0}^{t}F_{f}dt),$$
where $(v_{r}\int_{0}^{t}F_{f}dt$) represents the mechanical contribution in the *MRR* during planarizing/polishing, and *C_c_* (Å/min kJ) is referred to as a chemical contribution [23] in the *MRA* of the CMP.

Therefore, the multiplication between *E_f_* and abrasive concentration condition may be defined as the mechanical energy rate accompanying the CMP process. Thus, the *MRA* combined with the interaction between chemical and mechanical energy is expressed with Equation (6) and the ratio between the *MRA* and mechanical energy can be an index of chemical contribution (*C_c_*) in the CMP process, i.e., the *C_c_* would be affected by the mechanical condition of the CMP process.

## 4. Results and Discussion

### 4.1. Friction Characteristics in the CMP Process

The aim of the CMP process is to obtain the global planarization/polishing of the surfaces of the wafers (here covered with a selective layer), in which mechanical and chemical interactions are combined to achieve an efficient MRR. Mechanical interactions are manifested by *F_f_* and frictional heat in the CMP process. By introducing SiO_2_ as an abrasive in the selective-layer CMP slurry, both *F_f_* and MRR increase, and the calculated WNU is correlated with the COF. Additionally, the *F_f_* allows the calculation of the *E_f_* to facilitate the representation of mechanical action/abrasion during the CMP process.

Thus, Figure 1 shows the variation of *MRR* and *WNU* depending on *E_f_*. As shown in Figure 1, *E_f_* is related in particular to the *MRR*, not to the *WNU*. It is observed as *MRR* increases with the increasing *E_f_*. In the CMP process of the oxides, the input energy caused by the relative speed, the chemicals, and of contact is composed of energy, *E_f_*, thermal, *E_t_*, and vibration, *E_v_*. Among these components of energy, *E_f_* and *E_t_* participate at the *MRR*, and *E_v_* is dispersed in the surrounding environment [15,16]. According to Figure 1, the *MRR* is dependent on the *E_f_* in the CMP process; however, the *MRR* is related to *F_f_* through the quantity removed per unit length.

The factors that influence the *WNU* in the CMP process are largely divided into processing and equipment conditions, chemicals, and consumables. Therefore, the *WNU* depends on the CMP planarization mechanical characteristics in the work process if there are no changes in the consumables and the slurry composition. Experimentally, it turned out that there was a connection between the friction coefficient (*COF*) and the *WNU*. Thus, Figure 2 shows the evolution of the *MRR* and *WNU* depending on *COF*. The *COF* was introduced to characterize the friction effect and was determined by measuring *F_f_* and establishing the resultant normal force during the CMP process. According to Figure 2, as the *COF* increases, the *WNU* also grows, due to the resultant normal force’s position [13,20]. The resultant normal force is determined from the moment’s equilibrium equation relative to the planarization (polishing) head center, as it moves away from the planarization/polishing platen.

Figure 3 shows friction characteristics (friction energy, *E_f_*, and average friction force, *F_f_*) with the change of rotational speed, *n*(*v_r_*), and pressure, *p*.

Figure 3a shows the variation *E_f_* and average *F_f_* with various rotating speeds at constant pressure, *p*. While *n*(*v_r_*) increases, the *E_f_* also increases; however, the average *F_f_* decreases because the removed amount per unit time is prolonged with an increase in n(*v_r_*). Moreover, the decrease in *F_f_* with growing *n*(*v_r_*) also takes place due to the support provided by the boundary layer between the wafer and pad formed as a result of the suspension flow via the pressure dynamic, *p* [16]. Figure 3b shows the variation of *F_f_* and *E_f_* with various *p* of planarization/polishing at an *n*(*v_r_*) constant. With the increase in *p* of planarization/polishing, both the *F_f_* and *E_f_* grow due to the real contact area influence, caused by the *p* of planarization.

### 4.2. Thermal Effect in the CMP Process

The temperature generated by the friction to the pad–wafer contact in the CMP process is a critical parameter for the *MRR*. At the same time, this temperature is inevitable during the CMP process because it is provoked by *F_f_* due to the abrasive nanoparticles and the CMP slurry chemistry. Therefore, Figure 4 illustrates how the CMP process friction characteristics are dependent on the material mechanical and chemical removal. Thus, the friction is caused the most by the abrasive nanoparticles in the CMP slurry, as seen in Figure 4a. Additionally, it is observed that the CMP process temperature, depending on the pressure, *p* (see Figure 4b), generated through de-ionized water, is bigger than that that generated through the slurry. The de-ionized water was used in the experiments to significantly highlight the effect (influence) of the abrasive nanoparticles in the slurry, as seen in Figure 4. Therefore, the process temperature also contributes to the *MRR*, which is caused by *F_f_* in the CMP process.

Therefore, during the friction process in the CMP, the change in the CMP process temperature becomes essential. The effects of the slurry nanoparticle concentration and size were investigated by many researchers [16,24,25,26], often with contradictory results, because it influences the *MRR* during the CMP process.

This discrepancy explains itself based on the size concentration and the properties of the nanoparticles in the slurry and the planarization technique. At the same time, it indicates that the different planarization processes can become dominant for a certain planarization system as a mode of CMP application.

To solve the differences between the applied planarization rates, two material removal mechanisms were explained based on the silica (SiO_2_) CMP [2,8,15,27]. Figure 5 highlights the *MRR* distribution and *F_f_* obtained with 50 nm nanoparticle size SiO_2_ slurry depending on the slurry abrasive concentration (from 0.0 to 30 wt.%). At a low abrasive nanoparticle concentration in the slurries (0.0–7.5 wt.%), the *MRR* increases with growth in the concentration of abrasive nanoparticles, indicating the mechanical removal mechanism. The surface micrographs of the wafer obtained with an AFM from Figure 6 present the planarized surface in the base-slurry (Figure 6a) and scratches on the planarized surface in the slurry with SiO_2_ abrasive concentration of 15 wt.%, after the CMP process (Figure 6b).

The surface roughness of the wafer planarized in the base-slurry (see Figure 6a) was Ra of 1.512 nm, starting from Ra (before planarization) of 4409 nm, measured with a Portable Optical Profilometer (JR100, Nanovea SRL, European Office, Rivalta di Torino, Italy). This roughness of the wafer surface was measured after one hour of chemical–mechanical planarization.

The AFM micrograph (see Figure 6b) shows that the nanoparticles started sliding and rolling on the wafer as a result of the friction and the *MRR*. Additionally, the depth of the scratches caused by slurry SiO_2_ abrasive concentration was less than 7 nm.

It is notable that *MRR* reaches its maximum value at an abrasive nanoparticle concentration of 7.5 wt.%, after which a significant decrease occurs. This change is caused by the motion modification of the nanoparticles, and an increase in their number leads to the load per nanoparticle decreasing when in contact with the wafer surface. Then, for the abrasive concentrations between 7.5 and 15 wt.%, the nanoparticles start to roll faster rather than slide on the wafer surface [16]. The surface AFM micrograph at 15 wt.% shows pitting deformations rather than scratches on the wafer surface, demonstrating the rolling motion of the abrasive nanoparticles (Figure 6b). Therefore, at higher abrasive nanoparticle concentrations, *MRR* is primarily due to chemical interactions which significantly reduce the mechanical removal action.

Figure 7 describes the *MRR* depending on the concentration of abrasive nanoparticles (in the range 0.0–7.5 wt.%) with the 50 nm abrasive size. For planarization, a relative speed of 0.75 m/s, a pressure of 0.05 MPa, and a slurry flow of 150 mL/min were used. It can be observed that as the abrasive nanoparticle concentration increased, the *MRR* of the selective layer also increased. Thus, the *MRR* was about 85 nm/min when the abrasive nanoparticle concentration was 0.0 wt.%. This value is the mechanical removal/abrasion result between the pad asperities and the selective layer, respectively, by the selective-layer chemical dissolution. When 0.5 wt.% SiO_2_ was added in the selective-layer CMP slurry, the *MRR* reached approximately 132 nm/min. The *MRR* increased rapidly until the abrasive nanoparticle concentration of 2.5 wt.%, and then increased slowly as the abrasive nanoparticle concentration grew from 2.5 wt.% to 7.5 wt.%.

In support of these observations, in situ *F_f_* measurements showed an evolutionary trend of the total *F_f_*, similar to that of the *MRR* depending on the abrasive nanoparticle concentration in the slurry with the size of 50 nm (see Figure 5).

Additionally, it was proved that in situ measurements during the CMP process are correlated with the *MRR* response presented in ref. [7], i.e., the load per particle decreases with the increasing abrasive nanoparticle concentration, at which point these start rolling, leading to the reduction of *F_f_*.

Figure 8 shows the variation *F_f_* and planarization/polishing temperature with the planarization time in the CMP process, under the same conditions as those shown in Figure 7.

The friction force signals are presented in Figure 8a, and those of the process temperature in Figure 8b, depending on the planarization time. The friction appears in the early stage of planarization, because the *F_f_* is the effect of air pressure, respectively, of the relative movement between the wafer–pad when the static *F_f_* is exceeded. Additionally, the average *F_f_* increases with the growth of the abrasive nanoparticle concentration, changing from 18.50 N to 57.00 N. During the CMP, colloidal SiO_2_ is chemisorbed by the passivation selective layer, because the adhesion forces of the abrasives are usually about twice as large as the van der Waals forces due to the covalent bonds present [9]. Thus, the friction behavior and *MRR* in the CMP process could be influenced by the chemical absorption of SiO_2_ into the selective layer. The *F_f_* and *MRR* could also increase as the abrasive nanoparticle concentration increases as a result of the chemisorbed active abrasives number by the selective layer. As can be seen from Figure 8b, during the CMP process the planarization temperature increased while growing the concentration of abrasive nanoparticles. This could be related to the *E_f_*, because it would hasten the realization of a protective layer (soluble passivation layer) due to the chemical dissolution. Lee et al. in ref. [28] and Kawaguchi in ref. [29] confirmed that the inhibitor leads to the soluble passivation layer (protective layer) formation, indicating the selective-layer surface characteristic zone immersed in the slurry. Additionally, they demonstrated that the soluble layer of passivation is removed through the abrasive nanoparticle mechanical action because the protective layer covering the amount by inhibitor decreased on the selective-layer surface. As a result, the number of active abrasive nanoparticles has an effect on the growth of the *F_f_*; hence, the protective layer removal amount could also be increased.

The experimental results obtained in this paper during the CMP process were useful to examine the temperature distribution in the planarization pad. Additionally, a kinematic analysis of temperature distribution was performed to understand its growth, considering the friction characteristics. Therefore, the results of this examination can help in developing a more effective CMP process by estimating the temperature distribution in a planarization pad.

Figure 9 shows *MRR* and *E_f_* variation for a single abrasive depending on the abrasive concentration. As shown in Figure 9a, with the increased abrasive nanoparticle concentration, the *MRR* per single abrasive decreases. The probability of the active abrasive nanoparticle being in contact with the wafer, even by their number growth at a high concentration or with the pad asperities, is possible during the CMP process. It is proven that the *MRA* by using an abrasive nanoparticle is bigger than the *MRR* only by the pad asperities because the planarization pad has viscoelastic a property from being made of polyurethane. During the CMP, the pressure applied on the contact surface between a single abrasive and the wafer could be reduced, but due to the active abrasives number, the contact area increases, and a higher *MRA* could be obtained.

The average *E_f_* which is used for the characterization of friction behavior is time-dependent and cannot influence the shape of the friction curve. Hence, this characterization in the CMP process would be more reasonable with a singular *E_f_* because it depends on the planarization time and the friction curve shape. The *E_f_*, depending on abrasive concentration, for a single abrasive is plotted in Figure 9b, where it is observed that the *E_f_* decreases with the increasing abrasive nanoparticle concentration. Instead, the *E_f_* increases, because on a single abrasive the applied pressure increases if the active abrasive nanoparticle number decreases.

Figure 10 shows the *MRA* versus *E_f_* in different conditions of pressures and relative speeds that generate *F_f_* after the selective layer CMP process. It can be observed that by decreasing the abrasive nanoparticle concentration and increasing the *E_f_,* it is obtains higher *MRRs*.

When the *MRA* is plotted represented depending on *E_f_*, the curves show similar trends at different concentrations (see Figure 10), except that at the low values of *E_f_* (close to zero), *MRA* increased rapidly after 0.5 kJ and continued to grow relatively slower (with a slower rate). It seems that the *MRA,* depending on *E_f_*, is adequate to effectively characterize each concentration value of the abrasive nanoparticles, while the *MRA,* depending on *F_f_*, is useful (by the *F_f_* monitoring) to estimate *MRR* during CMP. Between the calculated *MRA* with Equation (5) and that experimentally determined, there were no significant differences, indicating that the theoretical and experimental results are in correlation, which proves (confirms) the planarization slurry stability in the CMP process. When the roughness and deformation of the surface are minimal, the surface quality is considered to be optimal. Therefore, the optimal unfolding of the CMP process presents importance for both nanoparticle–nanoparticle and nanoparticle–substrate interactions [30]. Thus, the surfactant presence can influence the lubrication between the abrasives and the planarization surface [28,29]; as a result, the *F_f_* can decrease, leading to the reduction of the *MRR*.

To this, the applied force calculation during planarization on a single abrasive was necessary [7]. This was possible by determining the contact area resulting from the applied pressure between the pad and the nanoparticle by the pad’s head [16], which is consistent with the modeling from previous studies [12,18,25,28]. Hence, it is possible for the *MRR* to be controlled by the lubrication effect introduced by the surfactant through the friction forces between the wafer and the abrasive nanoparticles. Additionally, studies with surfactants have shown that lateral forces exhibit nonlinear behavior with respect to normal forces (Figure 11).

Another approach for quantifying the friction forces that appear during the CMP process resulted from the observation that the *MRR* values compared with the baseline slurry were lower (see Figure 11). In-depth studies by simulation with AFM on the friction in situ between single-particle and the substrate in suspension with a surfactant highlighted the response of friction force. These simulations demonstrated that the slurry chemistry, including the composition, pH, and adhesion tension of the surfactant, could be adjusted to ensure the necessary friction force for optimal planarization performance [25,30]. A certain surface-active surfactant has an optimum concentration range, which ensures that the dispersion ability results in a better surface quality. Therefore, surfactants can be used to modify the nanoparticle–substrate and the nanoparticle–nanoparticle interactions in the CMP process, s, with the goal of optimizing it based on the selected/desired performance (surfactant and the optimum concentration range).

## 5. Conclusions

The frictional and thermal features of the selective-layer CMP process were analyzed in correlation with the *MRR* by the monitoring of *F_f_* depending on SiO_2_ concentration in an acid slurry mixed with an oxidizer, corrosion inhibitor, complexing agent, and surfactant.

The *F_f_* can decrease by increasing the planarization pad rotating speed at a constant pressure, and the slurry boundary layer support formed between the wafer and pad. Additionally, the *F_f_* can increase due to modification of the real contact area with increasing pressure, and *WNU* calculated from the balance moments equation in ratio with the planarization head center was in correlation with the *COF*.

The slurry SiO_2_ abrasive nanoparticles generate *F_f_* (based on which the *E_f_* is calculated/determined) and, implicitly, the CMP process temperature. The *E_f_* generated during CMP influences the modification of the CMP process temperature, presents importance in unfolding the chemical reactions and accelerates the *MRR* by the surface material activating, which is to be removed.

The colloidal SiO_2_ abrasive during the CMP helps remove the protective layer (passivation layer) by mechanical abrasion. By the addition of abrasive nanoparticles into the CMP slurry, the *MRA* of the selective layer has been improving, respectively; as the active abrasive nanoparticle concentration grows, it is possible that the *F_f_* and *MRR* increase due to chemisorbed nanoparticle number by the selective layer.

The *MRR* of the selective layer on the wafer surface into the CMP slurry, depending on *E_f_* after planarization with SiO_2_ concentrations, varied and had a non-linear characteristic, i.e., a non-linear (Prestonian) behavior. This behavior is useful to improve planarization level and wafer surface non-uniformity due to the various pressure distributions.

Therefore, in the CMP process, the *F_f_* participates in mechanical abrasion of the selective layer; instead, a large *F_f_* can generate defects, such as scratches and etching. It is very important to obtain a planarized surface of good quality and with optimal results, which is possible by controlling *F_f_*.

The oxide present on the metallic selective layer surface during CMP affects the selective-layer *MRR* (is responsible for *MRR*), respectively, and the layer oxidation rate and the oxide film solubility seem to condition the *MRR* of the selective layer for the slurry used.

For optimal results and the planarization/polishing performance, the slurry composition should be closely monitored during the CMP process, because the effect is significant.

The slurry’s chemical composition must be completely free of any other soft or hard nanoparticles (regardless of concentration) because it would lead to inconsistent *MRRs*, even with the deterioration of the surface quality. Thus, slurry stabilization in extreme conditions presents importance at the solid–liquid interface, and it is conducted by using surfactants. In this way, the CMP process is improved and has beneficial (positive) effects on the surface quality (minimal deformations and roughness).

At the same time, in the CMP process, the selective layer surface chemical state has an important role in the *MRR* of the layer.

To verify the chemical contribution in the selective layer CMP, an empirical model for the *MRA* was derived from the *E_f_* equation and validated by the CMP results.

Additionally, by reducing at a minimum the surface defects, with an optimal *MRR*, it can attain a quality planarization. This requirement can be met in the CMP process through abrasive nanoparticle stability and properties or by slurry chemistry engineering, respectively, by the control of the mechanical and chemical interactions.

## Figures and Tables

**Figure 1 materials-16-02550-f001:**
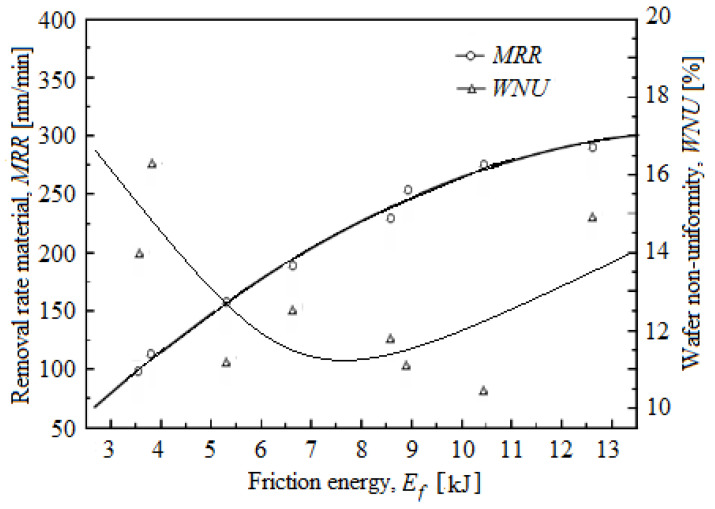
Removal rate material (*MRR*) and wafer non-uniformity (*WNU*), depending on friction energy (*Ef*).

**Figure 2 materials-16-02550-f002:**
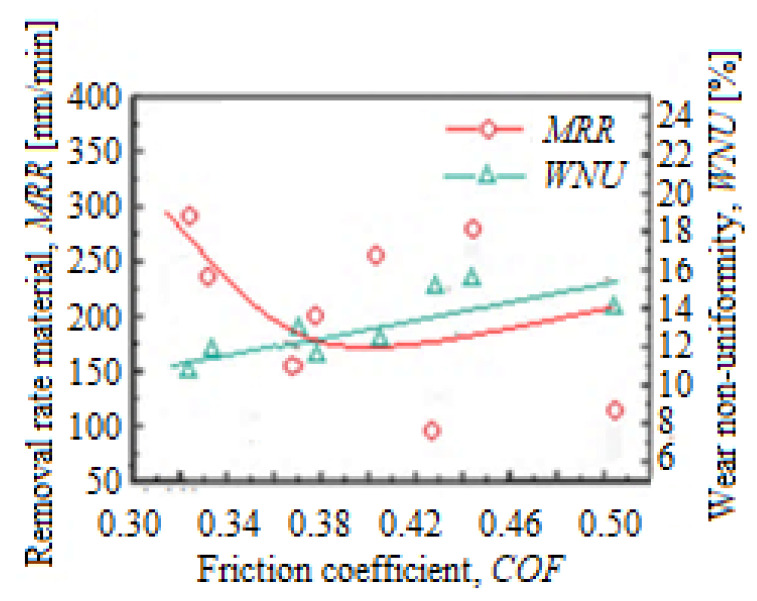
Variation of *MRR* and *WNU* depending on *COF* in the CMP process.

**Figure 3 materials-16-02550-f003:**
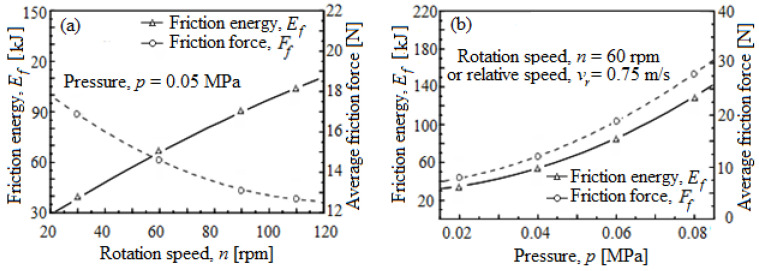
Variation friction characteristics (*E_f_* and *F_f_*) with the process parameters: (**a**) *E_f_* and average *F_f_* depending on *n*(*v_r_*), at *p* constant; (**b**) *E_f_* and average *F_f_* depending on *p* at *n*(*v_r_*), constant.

**Figure 4 materials-16-02550-f004:**
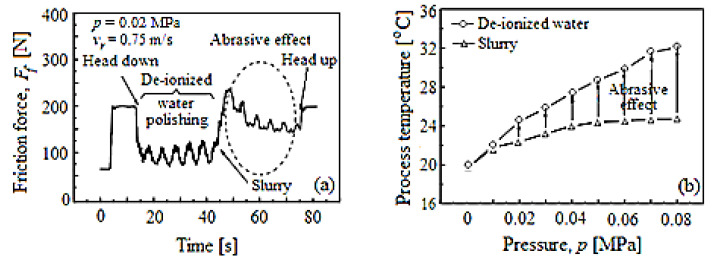
Abrasive nanoparticle’s effect on *F_f_* and process temperature during the CMP process: (**a**) the friction force, *F_f_*, using planarization with de-ionized water; (**b**) process temperature during planarization with de-ionized water and with CMP slurry, depending on the pressure, *p*.

**Figure 5 materials-16-02550-f005:**
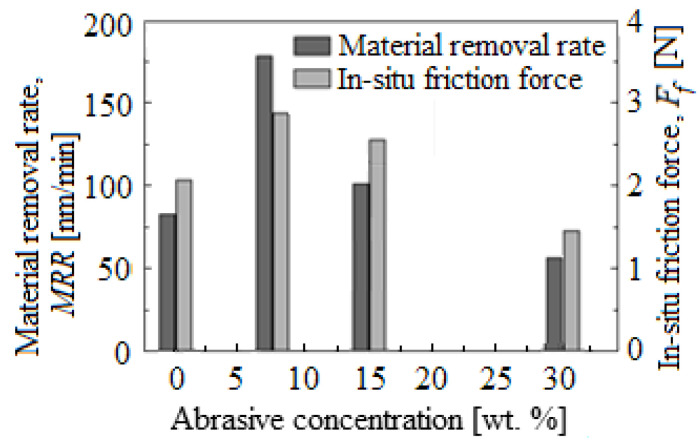
*MRR* and *F_f_* distribution made with the nanoparticle size of 50 nm in SiO_2_ slurry, depending on the slurry’s abrasive concentration.

**Figure 6 materials-16-02550-f006:**
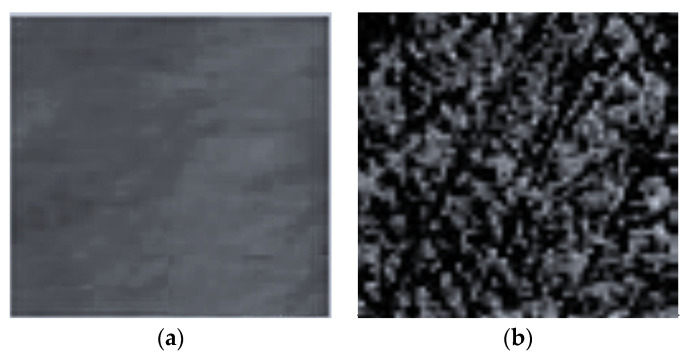
AFM micrographs of wafer surface planarization with: (**a**) base-slurry; (**b**) 15 wt.% slurry abrasive concentration and with sliding and rolling deformations on the wafer surface.

**Figure 7 materials-16-02550-f007:**
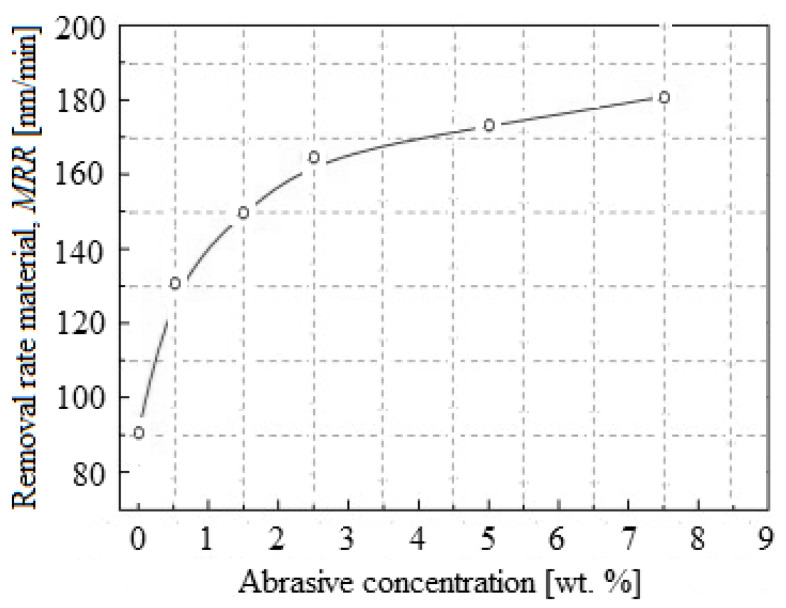
*MRR,* depending on the abrasive nanoparticle concentration.

**Figure 8 materials-16-02550-f008:**
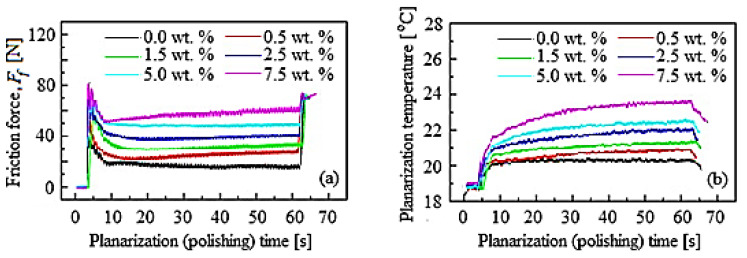
Friction force and planarization temperature, depending on the planarization time at the abrasive different concentrations: (**a**) friction force; (**b**) planarization temperature.

**Figure 9 materials-16-02550-f009:**
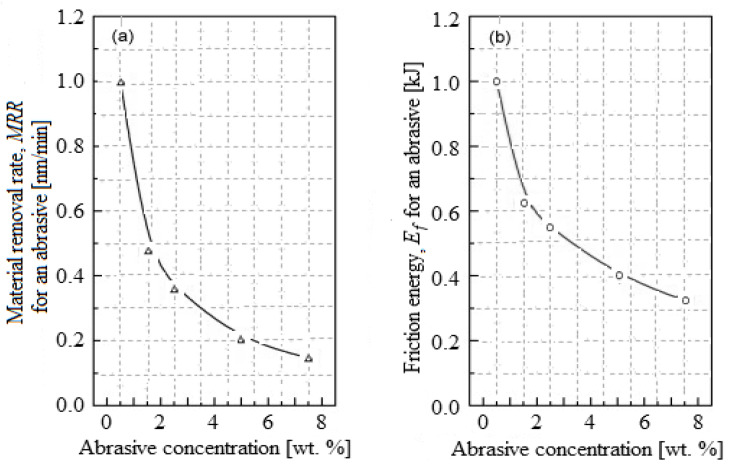
Friction variation of a single abrasive depending on abrasive concentration: (**a**) *MRR*; (**b**) *E_f_*.

**Figure 10 materials-16-02550-f010:**
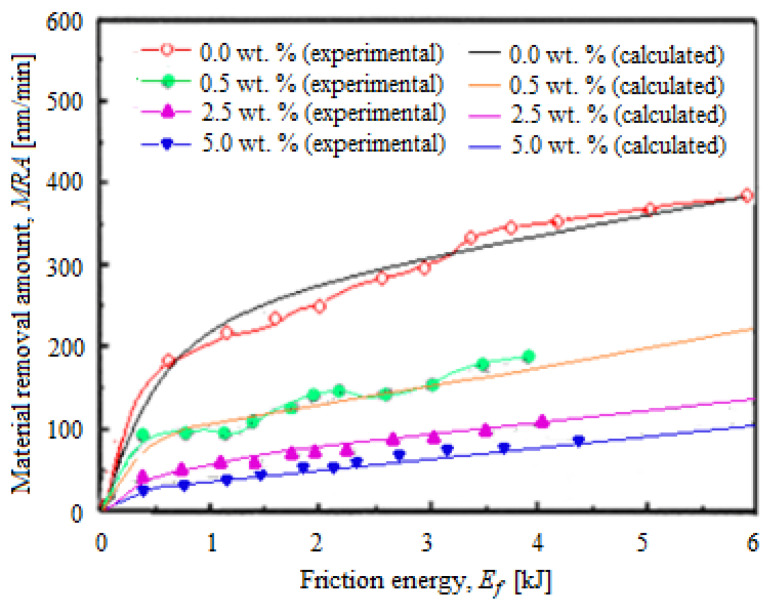
*MRA* variations after the CMP process under different pressures, relative speeds, and abrasive concentrations depending on the *E_f_*.

**Figure 11 materials-16-02550-f011:**
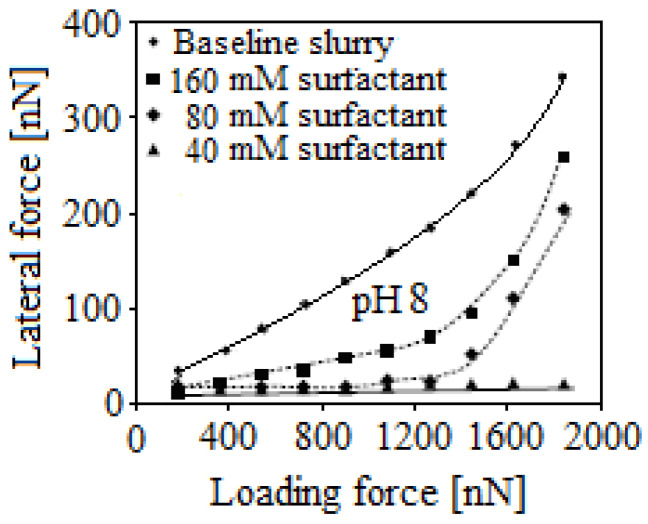
Friction force measurements on a selective-layer wafer in slurries with the pH8, a particle size of 50 nm, and surfactants at different concentrations, with the help of an AFM.

## Data Availability

Data are contained within the article.

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
