# Peer review of "The Effects of Friction and Temperature in the Chemical–Mechanical Planarization Process"

_materials, 2023, doi:10.3390/ma16072550_

Round 1

Reviewer 1 Report

The manuscript proposes an experimental study on the friction and thermal characteristics effect in the CMP process. It is novel that friction energy Ef was introduced in CMP to explain the friction effect. The influence of colloidal SiO,abrasive and other friction characteristics on the MRR of the selective layer are emphasized.

I recommend revisions before publication. More specific comments are below:

1. Suggest the authors review the manuscript for grammar problems of some words and sentences, which make the manuscript very difficult to understand.

2. In page 4 and lines 153-154, the authors concluded "the Ff has no relation to the MRR", why? This view breaks through the conventional views of tribology theory.

3. In page 4 and lines 161-167, COF is introduced to characterize the friction 

effect. How to get these COF values? In Fig. 2, the WNU points are irregularly distributed. Why does WNU depend on COF?

4. In page 5 and lines 185-194, why are the experiments not carried out between the slurries with or without abrasive nanoparticles, but using the de-ionized water instead?

5. In Fig. 10, Ef is taken as the horizontal axis. The Ef values seem to be 0.5, 1, 1.5, 2, 2.5, 3, 3.5, 4, and 5. However, the friction force Ff is shown in Fig. 8. According to Eq. (3), how to get these regular Ef values?

Author Response

Response Letter

for Reviewer 1, Round 1

Manuscript ID materials-2157637

          Entitled:

" The Effect of Friction and Thermal Characteristics in the CMP Process of the Selective Layer Surface"

                                                                            by

Filip Ilie, Ileana-Liliana Minea, Constantin Daniel Cotici,  Andrei-Florin Hristache

Note: The manuscript is very colorful. What was deleted is marked in red and cut, and what was entered is marked in green.

Thanks to Reviewer 1 for the comments made!

By the changes and additions realized I hope to be in the same consensus as the reviewer's wish!

Thus:

  1. Reviewer 1 says: Suggest the authors review the manuscript for grammar problems of some words and sentences, which make the manuscript very difficult to understand.

Response 1: Thank you for the suggestion and I inform you that the manuscript has been revised by a native English speaker and the grammatical problems have been corrected and I hope that the manuscript will be easy to understand! (see manuscript).

  1. Reviewer 1 says: In page 4 and lines 153-154, the authors concluded "the Ff has no relation to the MRR", why? This view breaks through the conventional views of tribology theory.

Response 2: Thank you for the observation because it is a completely wrong statement. As a result, it was corrected accordingly: what is marked in red and cut out was removed, and the completion is marked in green. So, „According to Figure 1, the MRR is dependent on the Ef in the CMP process; however, the Ff has no relation to the MRR, but it depends on the amount the MRR is related to Ff through the quantity removed per unit length” (see manuscript).

  1. Reviewer 1 says: In page 4 and lines 161-167, COF is introduced to characterize the friction effect. How to get these COF values? In Fig. 2, the WNU points are irregularly distributed. Why does WNU depend on COF?

Response 3: Thanks for the observation! Indeed COF was introduced to characterize the effect of friction. The COF was obtained by measuring the friction force in the CMP process, because above, in section 2 "Materials and Methods", it was specified "The CP-4 CMP planarizer is properly equipped with monitoring sensors in-situ, to signal (detected) the friction force and temperature during the CMP process. The signals emitted by sensors, after they were beforehand converted and amplified, were transferred to the CMP analysis software (a data acquisition system)." So, with the help of a sensor, found in the endowment of the used CP-4 CMP equipment, the friction force was measured during the CMP process and recorded in the data acquisition system. Knowing the friction force from the experiment and the normal force determined from the moment balance equation relative to the center of the planar head, as it moves away from the planar plate, the COF values shown in Figure 2 were obtained. How MRR is related and of the friction force, therefore implicitly also of the MRRavg that intervenes in relation (1), results in the dependence of WNU on COF, Although, the points in Fig. 2 are irregularly distributed, due to the change in the point of application of the normal force compared to the center of the planarization head, the curve between these points shows the evolution of WNU with COF. So, it has been experimentally proven that there is a connection between WNU and COF (see manuscript).

  1. Reviewer 1 says: In page 5 and lines 185-194, why are the experiments not carried out between the slurries with or without abrasive nanoparticles, but using the de-ionized water instead?

Response 4: Thank you for the question! The answer is simple; the experiments were made between the slurry with abrasive nanoparticles and the de-ionized water to significantly highlight the effect of the abrasive. If the experiments are carried out only between the slurries with or without abrasive nanoparticles, the effect of the abrasive was almost insignificant, because the difference is very small, and by graphic representation, the curves almost overlap.

As a result,  the sentence "Thus, the friction is caused the most by the abrasive nanoparticles in the CMP slurry, observable thing in Figures 4(a) and 4(b), it is observed that the generation CMP process temperature generated through de-ionized water is bigger than that generated through the slurry, depending on pressure, p." is split into two sentences,  corrected and completed, becoming "Thus, the friction is caused the most by the abrasive nanoparticles in the CMP slurry, observable thing in Figures 4(a) and 4(b). , it Also, it is observed that the generation CMP process temperature the CMP process temperature depending on the pressure, p (see Figure 4(b), generated through de-ionized water is bigger than that generated through the slurry. , depending on pressure, p. ", to which the sentence is added “The de-ionized water was used in the experiments to significantly highlight the effect (influence) of the abrasive in the slurry, as seen in Figure 4.”, followed by other grammatical corrections. (see the manuscript).

  1. Reviewer 1 says: In Fig. 10, Ef is taken as the horizontal axis. The Ef values seem to be 0.5, 1, 1.5, 2, 2.5, 3, 3.5, 4, and 5. However, the friction force Ff is shown in Fig. 8. According to Eq. (3), how to get these regular Ef values?

Response 5: Thank you for this observation which is correct! Indeed, checking the analytical and experimental results, they were represented erroneously. As you will notice, Figure 10 has been redone with the values obtained experimentally and analytically. Yes, in Figure 10, Ef is taken as the horizontal axis on which the values listed by you (usual) are passed and they are not values obtained analytically or experimentally, but are values on a scale, against which the values calculated analytically and compared with the experimental ones through the points represented in the figure and through which the MRA variation curves were drawn according to Ef (see the manuscript).

Note: With sincere apologies for these lapses and in the hope that now everything is in order, thank you once again for the observations and comments presented!

So, please you to review the entire manuscript, because more additions and grammatical corrections appeared!

Thanks for your understanding!

Date: January 16, 2023                                                                                         Authors

Reviewer 2 Report

(1) There are too many useless and repeated sentences in ‘Introduction’ and ‘Conclusion’,(e.g. line 29 – 37, line 411 - 414).

(2) The author should clarify the correlation between measured temperature and process temperature in ‘Experimental Details and Results’

(3) Figure 6 shows the scanning image of one specimen, however the AFM image and roughness measurement should be shown to illustrate the effect of silica’s concentration in the copper CMP process.

(4) In Figure 4 (b), CMP process in De-ionized water generated more heat than that in slurry with abrasive, which is contradictory to the statement by the authors that ’process temperature that is caused by Ff generated by the abrasive nanoparticles, in the CMP process’ in line 193-194.

(5) I wonder to know why the pressure is 0.05MPa which is much higher than that in industrial process or in other experiment.

(6) In Equation 5, the dimensions of MRA and MRR should be different. Material removal amount(MRA) seems to be the unit of total amount removed, while the material removal rate (MRR) is the unit of removal speed. Equation 5 and Y axis of Figure 10 should be reconsidered.

(7) Chemical reaction plays great role in CMP process. It should not be ignored when studying thermal effect in CMP process.

(8) The paper should by checked by a native speaker for improvement in language.

Author Response

Response Letter

for Reviewer 2, Round 1

Manuscript ID materials-2157637

          Entitled:

"The Effect of Friction and Thermal Characteristics in the CMP Process of the Selective Layer Surface"

                                                                            by

Filip Ilie, Ileana-Liliana Minea, Constantin Daniel Cotici,  Andrei-Florin Hristache

Note: The manuscript is very colorful. What was deleted is marked in red and cut, and what was entered is marked in green.

Thanks to Reviewer 2 for the comments made and I inform you that Manuscript ID materials-2157637 has been reviewed in accordance with your the observations, and recommendations and I will responded point by point to all your comments, explained as such and marked with green.

As proof, you will notice a lot of changes and additions and I hope to be in accordance with your requirements!

Thus:

  1. Reviewer 2 says: There are too many useless and repeated sentences in ‘Introduction’ and ‘Conclusion’,(e.g. line 29 – 37, line 411 - 414).

Response 1: Thanks for the observation! I apologize for the unnecessary sentences, which really exist and have been eliminated, marked in red and cut and completed and with other grammatical corrections (see manuscript).

  1. Reviewer 2 says: The author should clarify the correlation between measured temperature and process temperature in ‘Experimental Details and Results’

Response 2. I believe that by introducing the paragraph "Implicitly, the WNU calculation also demonstrates the non-uniformity of the CMP process temperature, so by measuring the temperature at the edge of the wafer-pad contact area, we cannot say with certainty that represents the real temperature. However, by measuring the temperature at several points on the edge of the wafer-pad contact area with the sensors help , which through the signals emitted and transferred to the data analysis software, we can consider with sufficient approximation that they are CMP process temperature characteristic."  I clarified the correlation between the measured temperature and the process temperature! (see manuscript).

  1. Reviewer 2 says: Figure 6 shows the scanning image of one specimen, however the AFM image and roughness measurement should be shown to illustrate the effect of silica’s concentration in the copper CMP process.

Response 3: Thanks for the observation and as you noticed, the AFM image of the wafer/specimen planarized for one hour has been inserted (Figure 6(a)), having the initial roughness (before planarization) Ra of 4,409 nm, because after planarization in the base-slurry, Ra was 1,512 nm. Also, corrections were made (marked in red and cut) and several sentences were added (marked in green), as follows: „The surface micrographs, in Figure 6 of the wafer (obtained with an AFM) present the planarized surface in the base-slurry (Figure 6(a)) and scratches on the planarized surface , at a in the slurry with SiO2 abrasive concentration of 15 wt. %, after the CMP process (Figure 6(b)).  Surface roughness of the wafer planarized in the base-slurry (see Figure 6(a)) was Ra of 1.512 nm, starting from Ra (before planarization) of 4,409 nm, measured with a Portable Optical Profilometer (JR100, Nanovea SRL, European Office, Rivalta di Torino, Italy). This roughness’s of the wafer surface was measured after one hour of chemical-mechanical planarization.

The AFM micrograph (see Figure 6(b)) shows that the nanoparticles started sliding and rolling on the wafer; having as a result of the friction and the MRR. Additionally, the depth of the scratches caused by slurry SiO2 abrasive concentration was less than 7 nm.”

 (see manuscript).

  1. Reviewer 2 says: In Figure 4 (b), CMP process in De-ionized water generated more heat than that in slurry with abrasive, which is contradictory to the statement by the authors that ’process temperature that is caused by Ff generated by the abrasive nanoparticles, in the CMP process’ in line 193-194.

Response 4: Thank you for this observation, correct by the way, which I have corrected, so that a contradiction no longer appears.. Thus, the sentence "Hence, the MRR contributes to the process temperature that is caused by Ff generated by the abrasive nanoparticles, in the CMP process." became "Hence, at the MRR contributes to and the process temperature that is caused by Ff generated by the abrasive nanoparticles, in the CMP process.”, to which some grammatical corrections were made, , as can be seen! (see manuscript).

  1. Reviewer 2 says: I wonder to know why the pressure is 0.05MPa which is much higher than that in industrial process or in other experiment.

Response 5: Thank you, you are right. With apologies for the rigor, it is a conversion error from psi to MPa, because in section 2 'Materials and Methods' it was specified that the applied pressurerom 0.5 psi to 3.0 psi! The considered pressure was 3 psi, which means approximately 0.02 MPa, which was corrected in Figure 4(a) (see manuscript).

  1. Reviewer 2 says: In Equation 5, the dimensions of MRA and MRR should be different. Material removal amount(MRA) seems to be the unit of total amount removed, while the material removal rate (MRR) is the unit of removal speed. Equation 5 and Y axis of Figure 10 should be reconsidered.

Response 6: Thank you for this observation, but equation 5 is correct both mathematically and dimensionally and dimensions of MRA and MRR are not different, although after the physical interpretation, it would be understood that way. In reality, both MRR and MRA are expressed with the same unit of measure nm/min (A/min in this paper) or can be in mg/min (like the wear intensity, which can be linear or gravimetric), because both represent removal/loss of material. Only that MRR is the removal rate of the material (after wafer/specimen) that is planarized/polished, and MRA is the total material loss/removal rate (both of the material/wafer to be planarized/polished and of the pad/tool that performs planarization/polishing operation). So, there should not be different units of measure. As you will notice in the manuscript (through several additions marked in green) and the introduction of equation (6), I hope this dilemma will be clarified. In addition, I think that the suggestion to reconsider equation (5) and the Y axis from Figure 10 is no longer the case and they can remain in their current form (without problems), because according to equation (6) and the explanations added in the manuscript, the MRA is expressed depending on Ef.

However, in Figure 10 the points marking the variation of MRA with Ef were reconsidered (see manuscript).

  1. Reviewer 2 says: Chemical reaction plays great role in CMP process. It should not be ignored when studying thermal effect in CMP process

Response 7: This statement is true, but the above answer (Response 6) does not ignore the chemical contribution in the CMP process, which is also taken into account in the MRA.

  1. Reviewer 2 says: The paper should by checked by a native speaker for improvement in language.

Response 8: I hope that now through all the grammatical corrections and additions requested by the reviewers, the English language has been properly improved. It should be mentioned that the work was checked by a native speaker and I tend to believe that I will not have to pay for the English language check again!

So, please you to review the entire manuscript, to see all grammatical corrections and additions!

Thanks for your understanding!

Date: January. 16, 2023                                                                                                      Authors

Author Response

Response Letter

for Reviewer 3, Round 1

Manuscript ID materials-2157637

          Entitled:

"The Effect of Friction and Thermal Characteristics in the CMP Process of the Selective Layer Surface"

                                                                            by

Filip Ilie, Ileana-Liliana Minea, Constantin Daniel Cotici,  Andrei-Florin Hristache

Note: The manuscript is very colorful. What was deleted is marked in red and cut, and what was entered is marked in green.

Thanks to Reviewer 3 for the comments made and I inform you that Manuscript ID materials-2157637 has been reviewed in accordance with your the observations, and recommendations and I will responded point by point to all your comments, explained as such and marked with green.

As proof, you will notice a lot of changes and additions and I hope to be in accordance with your requirements!

Thus:

  1. Reviewer 3 says: What kinds of wafers and what materials polished should be specified?

Response 1: Thank you for the indication and I inform you that in section 2 "Materials and Methods" the paragraph has been introduced "The friction and thermal characteristics investigation in the CMP process on a selective-layer surface was carried out on a wafer- pattern from metal with selective layer. The wafer-pattern from metal was made of OLC45 steel (equivalent AISI 1045) and previously coated on one face, with a selective layer (of thick, about 500 nm), by friction with the CuSn12T bronze (equivalent CC483K).",where the wafers and planarized materials are specified. (see manuscript).

  1. Reviewer 3 says: For this friction energy definition, it should be force*distance, not force*time. The reviewer cannot accept this definition, even though there is reference to quote.

Response 2. Thank you for your observation and I respect your choice. If you cannot accept this definition of Ef, although it exists in the references, but what you claim should be force*distance, not force*time, has support because in the relation of Ef intervenes vr, which is the ratio distance/ time (m/s), so force*distance/time appears in equation (3). As the friction force, Ff is variable in time (see Figure 8a), below the integral appears Ffdt (see manuscript).

Then,  Ef = , which is obtained by proceeding as follows:

The needed Ef as a single abrasive (ef) to scratch (planarize/polish) the surface of the selective layer wafer can be presented by the following equation:

,

where: f, s, vr and t - are the friction force acting on a single abrasive, space traversed by a single abrasive, relative velocity and polishing time, respectively.

It could be considered that the Ef applied on the wafer surface is the summation of the ef acting on the active abrasives which participate in the polishing and can be written can be written like this:                                                 ,

where: n and Ff - are the number of abrasives and friction force applied on the wafer surface by abrasives, respectively (see manuscript)

So, I hope that to some extent I have enlightened you, and you will agree with this definition of friction energy, Ef.

  1. Reviewer 3 says: If the polished material is specified, the paper title needs to be rephrased. The current title is too general.

Response 3: As I stated in Response 1, the planarized material is specified and can be found under the paper title "... of the Selective Layer Surface". In addition, the title of the paper has changed, as follows: 'The Effect of Friction and Thermal Characteristics in the CMP Process of the Selective Layer Surface' (see manuscript). 

  1. Reviewer 3 says: The figure quality should be improved to present in the paper.

Response 4: The figures quality has been improved, even some figures have been added (Figure 6a), changed (Figure 6b) or modified (Figure 10) (see manuscript).

So, please you to review the entire manuscript, to see all the additions!

Thanks for your understanding!

Date: January. 16, 2023                                                                                                       Authors

Reviewer 4 Report

Enough reservations are available throughout the manuscript to not to consider at this stage, following are point highlighted here and even other must be rectified in the revision:

The abstract needs to be revised based on the selected parameters and obtained results for CMP and further for MRR.

What is written from lines 29-36 in introduction?

I didn’t understand the last paragraph of the introduction, what authors want to address? Furthermore what is the importance and necessity to execute this work along with the novelty?

Which ASTM standard is used under heading 3?

How did you plot the results for Figure 4?

Figure 6 is relatively blurred; need to put a clarified image.

Is the Figured 10 is scanned or plotted graph, because the axes are titled, same query related t all the figures?

Author Response

Response Letter

for Reviewer 4, Round 1

Manuscript ID materials-2157637

          Entitled:

"The Friction and Thermal Characteristics Effect in the CMP Process of the Selective Layer Surface"

                                                                            by

Filip Ilie, Ileana-Liliana Minea, Constantin Daniel Cotici,  Andrei-Florin Hristache

Note: The manuscript is very colorful. What was deleted is marked in red and cut, and what was entered is marked in green.

Thanks to Reviewer 4 for the comments and observation made!

By the changes and additions realized I hope to be in the same consensus as the reviewer's wish, through the punctual and specific answers, highlighted by you!

Also, I hope that through all the corrections, changes and additions, all the reservations on the manuscript at this stage have been cancelled!

Thus:

  1. Reviewer 4 says: The abstract needs to be revised based on the selected parameters and obtained results for CMP and further for MRR.

Response 1: The abstract has been revised based on the selected parameters and the results obtained in the CMP process and I hope it will be in accordance with your observation! (see manuscript).

  1. Reviewer 4 says: What is written from lines 29-36 in introduction?

Response 2: Thank you for the observation, it's an oversight of ours (that's how it is when several people work, sometimes) and rows 29-36 were cancelled (see manuscript).

  1. Reviewer 4 says: I didn’t understand the last paragraph of the introduction, what authors want to address? Furthermore what is the importance and necessity to execute this work along with the novelty?

Response 3: Thank you for intiformation and I inform you that it has been corrected and completed, in order to understand the importance and novelty of this paper! (see manuscript).

  1. Reviewer 4 says: Which ASTM standard is used under heading 3?

Response 4: I inform you that the ASTM used has been included in the manuscript, namely: (ASTM F1530-94)  (see the manuscript).

  1. Reviewer 4 says: How did you plot the results for Figure 4?

Response 5: Figure 4 was taken from CMP analysis software (a data acquisition system), with which the CP-4 CMP planarization equipment is equipped and displayed on a monitor.

  1. Reviewer 4 says: 4. Figure 6 is relatively blurred; need to put a clarified image.

Response 4: Figure 6 has been redone and has two variants (a) and (b), because another reviewer requested it, and the variant considered unclear by you was changed  (see the manuscript).

  1. Reviewer 4 says: Is the Figured 10 is scanned or plotted graph, because the axes are titled, same query related t all the figures?

Response 5: Thank you for the question! This figure is not scanned but represented graphically. Since another reviewer had comments about the wrong presentation of the experimental and analytical points, it was corrected. And the other figures were represented graphically, photographed or taken from the CMP software (depends on the figure), (see manuscript).

Note: With sincere apologies for these lapses and in the hope that now everything is in order, thank you once again for the observations and comments presented!

So, please you to review the entire manuscript, because more additions and grammatical corrections appeared!

Thanks for your understanding!

Date: January 16, 2023                                                                                                      Authors

Round 2

Reviewer 1 Report

I suggest publication now.

Author Response

Response Letter

for Reviewer 1, Round 2

Manuscript ID materials-2157637

          Entitled:

"The Effect of Friction and Thermal Characteristics in the CMP Process of the Selective Layer Surface"

                                                                            by

Filip Ilie, Ileana-Liliana Minea, Constantin Daniel Cotici,  Andrei-Florin Hristache

I understand that by the changes, additions realized and the answers sent by the authors after round 1 satisfied the Reviewer 1, for which we thank you.

Thus:

  1. Reviewer 1 says: I suggest publication now.

Response 1: Many thanks to Reviewer 1 for suggesting the publication of the our paper.

Thanks for your understanding!

Date: January 22, 2023                                                                                                      Authors

Reviewer 3 Report

The basic concepts should be very clear.

Author Response

Response Letter

for Reviewer 3, Round 2

Manuscript ID materials-2157637

          Entitled:

"The Effect of Friction and Thermal Characteristics in the CMP Process of the Selective Layer Surface"

                                                                            by

Filip Ilie, Ileana-Liliana Minea, Constantin Daniel Cotici,  Andrei-Florin Hristache

Dear Reviewer 3,

You have put us in a delicate situation (we are in a big dilemma)! You didn't make any comments for the authors, but I simply noticed that you canceled what you checked in round 1.If after the 1st round no "Not applicable" box was checked (see the review of the 1st round), now, you have checked the 'Not Applicable' column, even though I have responded punctually to all your observations, comments or requests! What is the truth? We really don't understand and don't know what we should do! It's shocking!

So, please you to review the entire manuscript, to see all the additions!

Thanks for your understanding!

Date: January 22, 2023                                                                                                       Authors

Reviewer 4 Report

This article has some serious issues in the construction of heading, like heading No. 3. Experimental details and results can't be combined together, however, the experimental details can be linked with heading 2.

Results and discussion is a separate heading that provides the details of obtained results provided in the testing execution details in heading 2. So, this heading must be emerged as a separate heading.

Moreover, the conclusion section is completely vague, it seems like a story paragraph, which must highlight the conclusive bullets points of the results. 

Author Response

Response Letter

for Reviewer 4, Round 2

Manuscript ID materials-2157637

          Entitled:

"The Friction and Thermal Characteristics Effect in the CMP Process of the Selective Layer Surface"

                                                                            by

Filip Ilie, Ileana-Liliana Minea, Constantin Daniel Cotici,  Andrei-Florin Hristache

Note: The manuscript is very colorful. What was deleted is marked in red and cut, and what was entered is marked in green.

Thanks to Reviewer 4 for the comments and observations made!

With the changes and additions realized I hope to be in the same consensus with the reviewer, through the punctual and specific answers, highlighted by you!

Thus:

  1. Reviewer 4 says: This article has some serious issues in the construction of the heading, like heading No. 3. Experimental details and results can't be combined together, however, the experimental details can be linked with heading 2.

Response 1: Thanks for the observation! I took into account your suggestion and therefore heading No. 3 'Experimental details and results' has disappeared and a new heading 3 "Theoretical Aspects" has been introduced, which partially derives from the former heading 3, with additions marked with bleu color in text. In this case, it was no longer necessary to link to heading 2 (see manuscript).

  1. Reviewer 4 says: Results and discussion is a separate heading that provides the details of obtained results provided in the testing execution details in heading 2. So, this heading must emerge as a separate heading.

Response 2: Thanks for the suggestion! We also took this suggestion into account and introduced heading 4 "Results and Discussions" as a separate heading and contains part of the former heading 3 'Experimental details and results', where several paragraphs were canceled (marked in red and cut with a dash) and the additions are marked with blue color. (see manuscript).

  1. Reviewer 4 says: Moreover, the conclusion section is completely vague, it seems like a story paragraph, which must highlight the conclusive bullet points of the results.

Response 3: Thank you for your observation and you will notice that I have done everything possible so that the conclusions are no longer vague, but have been modified to highlight the salient points of the results (see manuscript).

Note: With sincere apologies for these lapses and in the hope that now everything is in order, thank you once again for the observations and comments presented!

Thanks for your understanding!

 Date: January 22, 2023                                                                                                      Authors

Round 3

Reviewer 3 Report

No.

Author Response

Response Letter

for Reviewer 3, Round 3

Manuscript ID materials-2157637

          Entitled:

"The Effects of Friction and Temperature in the Chemical-Mechanical Process"

                                                                            by

Filip Ilie, Ileana-Liliana Minea, Constantin Daniel Cotici,  Andrei-Florin Hristache

Dear Reviewer 3,

In addition, you have changed your opinion somewhat at "Open Review", but totally different in the three rounds, because in Round 1 you did not check any 'Not applicable', in Round 2 you checked the whole column 'Not applicable', for so that in Round 3 only three boxes with 'Not applicable' appear.

However, please notice that in Round 1, no box was checked for 'Not applicable', then in line 1 'Does the introduction provide sufficient background and include all relevant references?' 'Yes' was checked in Round 1, and in Round 3 'Not applicable'.

Also, in Round 1, the lines 'Are the results clearly presented?' and 'Are the conclusions supported by the results?' were marked with 'Must be improved', and in Round 3, with 'Not applicable'.

What is the truth? It means that the manuscript was better in Round 1! And after so many completions, corrections, and cancellations of sentences, may the manuscript be more inadequate! How can you have such different opinions, in such a short time, about the same issue?

Thanks for your understanding!

Date: February 06, 2023                                                                                                       Authors

Reviewer 4 Report

The last paragraph seems not fine in the introduction which highlights the novelty and the precise work of the authors.

This needs to be double check that the graphs are plotted on any software not the scanned images like Fig. 2, 3, 4, 5, 8, 9, 10. Kindly confirm this.

I think don't understand what Fig. 6(a) is representing.

Authors still need to check every section for major tuning of the article.

Author Response

Response Letter

for Reviewer 4, Round 3

Manuscript ID materials-2157637

          Entitled:

"The Effects of Friction and Temperature in the Chemical-Mechanical Process"

                                                                            by

Filip Ilie, Ileana-Liliana Minea, Constantin Daniel Cotici,  Andrei-Florin Hristache

Note: The manuscript is very colorful. What was deleted is marked in red and cut, and what was entered is marked in blue (after Round 1), green (Round 2), and light purple (Round 3).

Thanks to Reviewer 4 for the comments and observations made!

With the changes and additions realized I hope to be in the same consensus with you, through the punctual and specific answers, highlighted by you!

Thus:

  1. Reviewer 4 says: The last paragraph seems not fine in the introduction which highlights the novelty and the precise work of the authors.

Response 1: This observation is true and for which I thank you, as a result, the last paragraph in the introduction was canceled and replaced with the sentence "The results of this investigation can help in developing a CMP process more effectively, with an optimal MRR, to achieve a quality planarization." (see manuscript).

  1. Reviewer 4 says: This needs to be double check that the graphs are plotted on any software not the scanned images like Fig. 2, 3, 4, 5, 8, 9, 10. Kindly confirm this.

Response 2: Thank you for the suggestion and I inform you that the listed figures are graphically represented in 'Paint' or taken directly from the computer, which is why after they are transferred into the text of the work (or anywhere) they appear in a slightly different form than in 'Paint' (from as if they were scanned).

  1. Reviewer 4 says: I think don't understand what Fig. 6(a) is representing.

Response 3: This Figure 6(a) was inserted later, because one of the reviewers asked to specify the size of the wafer roughness before introducing abrasive nanoparticles of SiO2. Thus, Figure 6(a) represents the planarized wafer in the base suspension (without abrasive SiO2 nanoparticles) with Ra of 1,512 nm.

Thank you and I hope it is now understood!

  1. Reviewer 4 says: Authors still need to check every section for major tuning of the article.

Response 3: Thank you for the suggestion and please notice that the entire work has been checked, as a result there have been several corrections, additions, even sentences, marked in light purple color (see the manuscript)

Note: With sincere apologies for these lapses and in the hope that now everything is in order, thank you once again for the observations and comments presented!

Thanks for your understanding!

 Date: February 06, 2023                                                                                                       Authors
